# On Verifying Linear Execution Strategies in Planning Against Nature

**Primary Keywords:** *None*

## Abstract

While planning and acting in environments in which nature can trigger non-deterministic events, the agent has to consider that the state of the environment might change without its consent. Practically, it means that the agent has to make sure that it eventually achieves its goal (if possible) despite the acts of nature. In this paper, we first formalize the semantics of such problems in Alternating-time Temporal Logic, which allows us to prove some theoretical properties of different types of solutions. Then, we focus on linear execution strategies, which resemble classical plans in that they follow a fixed sequence of actions. We show that any problem that can be solved by a linear execution strategy can be solved by a particular form of linear execution strategy which assigns *wait-for preconditions* to each action in the plan that specifies when to execute that action. Then, we propose a sound algorithm that verifies a sequence of actions and assigns wait-for preconditions to them by leveraging abstraction.

## Introduction

Planning and acting in real-world scenarios (Ingrand and Ghallab 2017), such as planetary rovers (Ai-Chang et al. 2004), or autonomous underwater vehicles (Chrpa et al. 2015), poses a challenge as during the plan execution the environment might change by exogenous events that are not under the control of the agent. Exogenous events are triggered by an actor that does not have specific intentions (or goals) and acts rather randomly. We call that actor *nature*. The (rational) agent that wants to achieve its goal has to take into consideration possible actions (events) of nature. We call such a problem *planning against nature*. Acts of nature, however, might render the plan invalid, make the agent's goal no longer achievable, or, worse, they might cause damage to the agent.

For example, the *AUV domain* (Chrpa, Gemrot, and Pilát 2020), which is inspired by real-world AUV operations (Chrpa et al. 2015), simulates a scenario in which the AUV has to perform sampling of given objects of interest while there might be ships passing by in corridors that might endanger the AUV. The movement of the ship can be represented by exogenous events. If a ship enters the AUV's location, then the AUV is destroyed.

The concept of planning against nature is not new (Dean and Wellman 1990; Musliner, Durfee, and Shin 1993; Ioc-

chi, Nardi, and Rosati 2000) and usually requires to reason with the whole state space or a large portion of it. The problem can be modelled as a Fully-observable non-deterministic (FOND) planning task, which considers actions with non-deterministic effects, i.e. when an action is applied, the result of its application might have different outcomes (Cimatti et al. 2003). FOND planning is known to be EXPTIME-complete (Littman, Goldsmith, and Mundhenk 1998). However, the number of non-deterministic alternatives (per action) might even be exponential with respect to the number of events if we consider that nature can apply multiple events at once.

Leveraging classical planning techniques such as FF-replan (Yoon, Fern, and Givan 2007) (an unofficial winner of the probabilistic track of the International Planning Competition 2006), where if the agent is in an unexpected state it re-plans, is a promising alternative but does not give any guarantees of success. Recent works (Chrpa, Gemrot, and Pilát 2020; Chrpa, Pilát, and Med 2021) studied under which conditions a sequential plan is guaranteed to eventually succeed. In contrast to those works, we consider that nature can apply sequences of events at once and provide a formal definition of wait-for preconditions, inspired by the work on social laws in planning (Karpas, Shleyfman, and Tennenholtz 2017), that establish when a given action (in the sequence) can be applied. For example, consider an AUV navigating from west to east, which must cross the path of a ship moving from north to south. The same sequence of actions for the AUV might lead to different outcomes, depending on the timing – entering the corridor of the ship after the ship crosses guarantees the AUV will reach the goal while entering the path of the ship before the ship crosses might lead to a collision. Hence, the wait-for precondition should prohibit the latter case, i.e., the AUV has to wait until the ship passes.

Inspired by the line of work on Situation Calculus (Reiter 1996; Giacomo and Lespérance 2021), in this paper, we formalise planning against nature tasks by using a concurrent game structure and Alternating-time Temporal Logic (ATL) (Alur, Henzinger, and Kupferman 2002). Hence ATL model checking can be leveraged to generate execution strategies. We then focus on *linear execution strategies*, resembling classical sequential plans, where we show that the fundamental principle determining whether a sequence of actions can yield a linear execution strategy is to guarantee

that the next action (or the goal after the whole sequence is executed) becomes applicable (or the goal is achieved) after an infinite number of "turns" of nature. Also, we show that if we have a sequence of actions that can yield a linear execution strategy that guarantees achieving the goal, then we can compute *wait-for preconditions* for each action by stepwise regression from the goal that makes such a linear execution strategy unique. Then, we propose an algorithm that verifies a sequence of actions (generated by classical planners) by using a heuristic approach based on problem abstraction. On top of that, the verification algorithm computes wait-for preconditions for actions in that sequence. Although the proposed algorithm is theoretically incomplete, it allows to verify linear execution strategies generated by off-the-shelf classical planners for a subclass of problems in the complexity of classical planning, i.e., PSPACE-complete (Bylander 1994). We show empirically that our approach has the potential to solve a subclass of planning against nature problems.

# Preliminaries

**Planning against Nature** can be understood as a special case of multi-agent planning (Brafman and Domshlak 2008) in which an intelligent agent that plans towards its goal acts against a "random" agent (or nature) that acts randomly without a specific purpose (or goal).

To represent the environment, we use Finite Domain Representation (FDR) (Helmert 2009). Let $V$ be a set of ***variables*** where each variable $v \in V$ is associated with its domain $D(v)$. An ***assignment*** of a variable $v \in V$ is a pair $(v, val)$, where its value $val \in D(v)$. Hereinafter, an assignment of a variable is also denoted as a ***fact***. A (partial) ***variable assignment*** $p$ over $V$ is a set of assignments of individual variables from $V$, where $vars(p)$ is a set of all variables in $p$ and $p[v]$ represents a value of $v$ in $p$. A ***state*** is a complete variable assignment (over $V$). We say that a (partial) variable assignment $q$ ***holds*** in a (partial) variable assignment $p$, denoted as $p \models q$, if and only if $vars(q) \subseteq vars(p)$ and for each $v \in vars(q)$ it is the case that $q[v] = p[v]$.

An ***action*** is a pair $a = (pre(a), eff(a))$, where $pre(a)$ is a partial variable assignment representing $a$'s precondition and $eff(a)$ is a partial variable assignment representing $a$'s effects. We say that an action $a$ is ***applicable*** in state $s$ if and only if $s \models pre(a)$. The ***result*** of applying $a$ in $s$, denoted as $\gamma(s, a)$, is a state $s'$ such that for each variable $v \in V$, $s'[v] = eff(a)[v]$ if $v \in vars(eff(a))$ while $s'[v] = s[v]$ otherwise. If $a$ is not applicable in $s$, $\gamma(s, a)$ is undefined. The notion of action application can be extended to sequences of actions, i.e., $\gamma(s, \langle a_1, \ldots, a_n \rangle) = \gamma(\ldots \gamma(s, a_1) \ldots, a_n)$. We denote as $ha(a)$ a (partial) variable assignment representing values of variables that must ***hold after applying*** an action $a$, i.e., for each $v \in vars(eff(a)) : ha(a)[v] = eff(a)[v]$ and for each $v' \in vars(pre(a)) \setminus vars(eff(a)) : ha(a)[v'] = pre(a)[v']$.

We define a ***planning task against nature*** (or ***planning task***, for short) as a tuple $\mathcal{P} = (V, A, E, I, G)$, where $V$ is a set of variables, $A$ a set of actions of the agent, $E$ is a set of actions of nature (or, ***events***), $I$ a complete variable assignment representing the initial state and $G$ a partial variable assignment representing the goal.

**Alternating-time Temporal Logic** (Alur, Henzinger, and Kupferman 2002), abbreviated as ATL, is a modal logic which allows us to write formulas which describe interactions between multiple agents, and what a set of agents can achieve regardless of what the other agents choose to do. The semantics of ATL rely on the definition of a concurrent game structure:

**Definition 1.** *A **concurrent game structure** is a tuple* $\mathcal{S} = \langle P, S, F, \pi, d, \delta \rangle$ *with the following components:*
- *A set of players* $P = \{p_1, \ldots, p_k\}$
- *A finite set of states* $S$
- *A finite set of propositions* $F$ *(also called observables)*
- *For each state* $s \in S$, $\pi(s) \subseteq F$ *is a set of propositions which are true in* $s$
- *For each player* $p \in P$ *and each state* $s \in S$*, a set of moves* $d_p(s)$ *available to player* $p$ *in state* $s$*. Given a state* $s \in S$*, we write* $M(s)$ *for the set* $(d_{p_1}(s) \times \cdots \times d_{p_k}(s))$ *of* move vectors*. The function* $M$ *is called* move function.
- *For each state* $s \in S$ *and each move vector* $\langle j_{p_1} \ldots, j_{p_k} \rangle \in M(s)$*, the state* $\delta(s, j_{p_1}, \ldots, j_{p_k}) \in S$ *that results from state* $s$ *if every player* $p_i \in P$ *chooses move* $j_{p_i}$*. The function* $\delta$ *is called* transition function.

**Definition 2.** *Given a concurrent game structure, an ATL formula* $\Phi$ *is either:*
- *A single proposition* $f$ *for any proposition* $f \in F$
- *Composed of smaller ATL formulas* $\Phi_1, \Phi_2$ *using the propositional logical connectives:* $\neg \Phi_1$ *or* $\Phi_1 \vee \Phi_2$
- *Composed of smaller ATL formulas using a path quantifier* $\langle\langle \mathcal{A} \rangle\rangle$ *(where* $\mathcal{A} \subseteq P$ *is a set of players) and a temporal operator (* $\circ$ *(next),* $\square$ *(always), or* $\mathcal{U}$ *(until)):* $\langle\langle \mathcal{A} \rangle\rangle \circ \Phi_1$, $\langle\langle \mathcal{A} \rangle\rangle \square \Phi_1$, *or* $\langle\langle \mathcal{A} \rangle\rangle \Phi_1 \mathcal{U} \Phi_2$

The interpretation of the propositions ($F$) and logical connectives ($\wedge, \vee, \neg$) is straightforward, and the temporal operators — $\circ$ (next), $\diamond$ (eventually), $\square$ (always), $\mathcal{U}$ (until) — are similar to those used in LTL (Pnueli 1977).

The path quantifiers allow ATL to express properties of multi-agent systems. For a set of players $\mathcal{A}$, the formula $\langle\langle \mathcal{A} \rangle\rangle \Phi$, means that the players in $\mathcal{A}$ can ensure the formula $\Phi$ holds, regardless of what the other players ($P \setminus \mathcal{A}$) do. To define this formally, we define a strategy $z_p$ for player $p$ as a function that maps every sequence of states $\lambda \in S^+$ ending in state $s$ (that is, a possible history) to an action applicable in $s$, i.e., $z_p : \lambda \rightarrow d_p(s)$. Given a state $s$, a set of players $\mathcal{A}$, and a set of strategies $Z_{\mathcal{A}}$ (one for each player in $\mathcal{A}$), $out(s, Z_{\mathcal{A}})$ is the set of possible trajectories which could occur when starting from state $s$ and the players in $\mathcal{A}$ follow strategies $Z_{\mathcal{A}}$. In other words, a trajectory $s_0, s_1, \ldots, s_m \in out(s, Z_{\mathcal{A}})$ iff $s_0 = s$, and for $i \in \{0, \ldots, m\} \, \exists \langle j_1, \ldots, j_n \rangle \in M(s_i)$ such that $\delta(s_i, j_1, \ldots, j_n) = s_{i+1}$ and for $p_k \in \mathcal{A} : z_{p_k}(s_i) = j_k$. We can now define when an ATL formula is satisfied.

**Definition 3.** *Given a concurrent game structure* $\mathcal{S}$*, state* $s$*, and ATL formula* $\Phi$*, we say that* $\mathcal{S}, s \models \Phi$*. For brevity we omit* $\mathcal{S}$*, and define this recursively by:*
- $s \models f$ *iff* $f \in \pi(s)$
- $s \models \neg \Phi$ *iff* $s \not\models \Phi$

- $s \models \Phi_1 \vee \Phi_2$ *iff* $s \models \Phi_1$ *or* $s \models \Phi_2$
- $s \models \langle\langle \mathcal{A} \rangle\rangle \circ \Phi$ *iff there exists a set of strategies* $Z_{\mathcal{A}}$ *(one for each player in $\mathcal{A}$) such that for all trajectories* $s_0, s_1, \ldots s_j \in out(s, Z_{\mathcal{A}})$, $s_1 \models \Phi$
- $s \models \langle\langle \mathcal{A} \rangle\rangle \Box \Phi$ *iff there exists a set of strategies* $Z_{\mathcal{A}}$ *(one for each player in $\mathcal{A}$) such that for all trajectories* $s_0, s_1, \ldots s_j \in out(s, Z_{\mathcal{A}})$, *for all positions* $i \in \{0, \ldots, j\}$, $s_i \models \Phi$
- $s \models \langle\langle \mathcal{A} \rangle\rangle \Phi_1 \mathcal{U} \Phi_2$ *iff there exists a set of strategies* $Z_{\mathcal{A}}$ *(one for each player in $\mathcal{A}$) such that for all trajectories* $s_0, s_1, \ldots s_j \in out(s, Z_{\mathcal{A}})$, *there exists a position* $i \in \{0, \ldots, j\}$ *such that* $s_i \models \Phi_2$, *and for all positions* $0 \leq j < i : s_j \models \Phi_1$

Finally, we can discuss ATL fairness constraints. While there are several types of fairness discussed in the original ATL paper (Alur, Henzinger, and Kupferman 2002), we review only strong fairness constraints, which we use later.

**Definition 4.** *Given a concurrent game structure* $\mathcal{S} = \langle P, S, F, \pi, d, \delta \rangle$, *a fairness constraint is a pair* $\langle p, \lambda \rangle$ *where* $p \in P$ *is a player and $\lambda$ maps every state* $s \in S$ *to a set of moves in* $d_p(s)$.

*A trajectory* $s_0, s_1, \ldots$ *is strongly* $\langle p, \lambda \rangle$*-fair iff there are only finitely many positions* $i$ *where* $\lambda(s_i) \neq \emptyset$, *or there are infinitely many positions* $i$ *where* $\langle j_1, \ldots, j_n \rangle \in M(s_i)$, $\delta(s_i, j_1, \ldots, j_n) = s_{i+1}$, *and* $j_p \in \lambda(s_i)$.

In other words, a fairness constraint specifies a subset of moves, each of which must be taken infinitely often when each state is visited infinitely often. Note that it is similar to fairness required by strong cyclic FOND planning (Cimatti et al. 2003).

## Execution Model as a Concurrent Game

In the presence of non-deterministic events, we look for an execution strategy of an agent that, in a nutshell, specifies which action (if any) is applied in a particular state whenever the agent can act. Our execution model is derived from the model used in previous work on social laws (Karpas, Shleyfman, and Tennenholtz 2017) to account for possibly different durations of actions and events even without explicitly specifying those in the action/event description. Note that this model differs from the work on planning with events (Chrpa, Pilát, and Med 2021) that considered alternating turns of agent and nature (like in a game of chess). To resolve potential conflicts between actions and events, we consider in our execution model that sequences of events triggered by nature alternate with actions of the agent.

In the context of ATL, we can specify a concurrent game structure for two players – an *agent* and *nature* – where the agent is responsible for applying actions with the aim of achieving its goal while nature applies events randomly without a specific aim (we consider the fairness assumption, so each event has a chance to occur if it is applicable). We assume that each player can act only if it is its turn. Note that we do not consider a specific "scheduler" player for changing turns, as suggested in some work on ATL (Alur, Henzinger, and Kupferman 2002), but both players can switch turns by actions or a specific "switch" events.

**Definition 5.** *Let* $\mathcal{P} = (V, A, E, I, G)$ *be a planning task. We define a variable turn with domain* $D(turn) = \{at, nt\}$. *Then, we define an action* $switch_a = (\{(turn, at)\}, \{(turn, nt)\})$, *an event* $switch_e = (\{(turn, nt)\}, \{(turn, at)\})$ *and an empty action/event "0". Let $F$ be the set of all facts over* $V \cup \{turn\}$, $S$ *be the set of states over* $V \cup \{turn\}$.

*We define the* **concurrent game structure for** $\mathcal{P}$ *to be* $\mathcal{S} = \langle P, S, F, \pi, d, \delta \rangle$, *where:*
- $P = \{agent, nature\}$
- $\pi : S \rightarrow 2^F$ *with* $\pi(s) = \{(v, val) \mid v \in V, val \in D(v), s[v] = val\}$
- *for each* $s \in S$ *it holds that* $d_{agent}(s) = \{a \mid a \in A \cup \{switch_a\}, s \models pre(a) \wedge (turn, at)\} \cup \{0\}$, $d_{nature}(s) = \{e \mid e \in E \cup \{switch_e\}, s \models pre(e) \wedge (turn, nt)\} \cup \{0\}$
- $\forall a \in A \cup \{switch_a\}$ *s.t.* $s \models pre(a) \wedge (turn, at)$: $\delta(s, a, 0) = (\gamma(s, a) \setminus \{(turn, at)\} \cup \{(turn, nt)\})$,
- $\forall e \in E \cup \{switch_e\}$ *s.t.* $s \models pre(e) \wedge (turn, nt)$: $\delta(s, 0, e) = \gamma(s, e)$

*We define a fairness constraint* $(nature, \lambda)$ *such that* $\lambda(s) = d_{nature}(s)$ *for each* $s \in S$, *that is, nature will take each move infinitely often.*

The concurrent game structure models the interaction between the agent and nature. Although our model does not resemble a typical concurrent game, i.e., the agent and nature do not act simultaneously, it captures the interaction in an asynchronous way, where nature can apply a sequence of events before switching the turn back to the agent.

We can define an *execution strategy* for the agent as a function that for each state in which the agent has its turn maps an action that the agent applies in that state (including *switch*). The outcome of an execution strategy is a set of all possible sequences of actions that might occur when the agent executes the strategy.

**Definition 6.** *Let* $\mathcal{P} = (V, A, E, I, G)$ *be a planning task and* $\mathcal{S} = \langle P, S, F, \pi, d, \delta \rangle$ *be its concurrent game structure. Let $\sigma$ be a policy of the agent in the concurrent game structure $\mathcal{S}$, which maps every state* $s \in S$ *to an action* $d_a(s)$. *We define the* **execution strategy** $\epsilon_\sigma : S \rightarrow A \cup \{switch_a\}$ *by* $\epsilon_\sigma(s) = \sigma(s)$.

Note that by definition of $\mathcal{S}$, for every $s \in S$ such that $s[turn] = at$ it is the case that $\epsilon(s) = a \rightarrow s \models pre(a)$.

The notion of *outcome* of an execution strategy, formally introduced below, describes a specific play of the "game" specified by the concurrent game structure ($\mathcal{S}$) of a planning task ($\mathcal{P}$).

**Definition 7.** *An* **outcome** *of an execution strategy $\epsilon$ is a sequence of actions from* $A \cup \{switch_a\}$ *that the agent applies in the game (described by $\mathcal{S}$) starting in* $I \cup \{(turn, nt)\}$ *and, possibly, reaching a state* $s \models G \wedge (turn, at)$ *if the outcome is* **successful**.

Note that a successful execution strategy can reach the goal and then continue. If we want to stop at the goal we can either consider versions of ATL on finite traces (Belardinelli et al. 2018), or modify the definition of the concurrent game structure to include "noop" action for the agent at the goal, allowing the agent to loop forever once it reaches the goal.

However, for the sake of simplicity, we use the standard version of ATL, which is defined on infinite traces.

We can now define a *valid* execution strategy as a strategy whose all possible outcomes are successful, in other words, no matter how nature plays (if following the fairness assumption), the agent always reaches its goal by a valid execution strategy. We also say that a valid execution strategy is a *solution* of the given planning task.

**Definition 8.** *Let* $\mathcal{P} = (V, A, E, I, G)$ *be a planning task and* $\epsilon$ *be an execution strategy. If every outcome of* $\epsilon$ *is successful, then we say that* $\epsilon$ *is* **valid***. We also say that if* $\epsilon$ *is valid, then it is a* **solution** *of* $\mathcal{P}$.

We can reasonably assume that nature always switches the turn to the agent in a finite number of steps. Then, we can say that a planning task is solvable if and only if the agent can extract an execution strategy that eventually achieves the goal. The following definition captures the above aspects in an ATL formula.

**Proposition 1.** *Let* $\mathcal{P} = (V, A, E, I, G)$ *be a planning task,* $\mathcal{S}$ *be the concurrent game structure for* $\mathcal{P}$ *and turn be a variable as in Def. 5. We say that* $\mathcal{P}$ *is* **solvable** *if and only if* $\mathcal{S}, I \cup \{(turn, nt)\} \models (\langle\langle nature \rangle\rangle \Box ((turn, nt) \rightarrow \diamond (turn, at))) \rightarrow (\langle\langle agent \rangle\rangle \diamond (G \wedge (turn, at)))$.

*Proof.* If the premise $I \cup \{(turn, nt)\} \models (\langle\langle nature \rangle\rangle \Box ((turn, nt) \rightarrow \diamond (turn, at)))$ holds, and the formula above is satisfiable for $\mathcal{S}$, then by the definition of ATL semantics for $\langle\langle agent \rangle\rangle \diamond (G \wedge (turn, at)))$, there exists some strategy for the agent, such that for every strategy of nature, the goal $G$ is achieved for every outcome, i.e., the execution strategy of the agent is valid and a solution of $\mathcal{P}$. The "if" implication also straightforwardly holds as the existence of an execution strategy achieving $G$ implies the satisfiability of the ATL formula. $\square$

The above proposition indicates that a valid execution strategy can be extracted by model checking of the given ATL formula. Of course, ATL model checking is not computationally tractable – in fact, with strong fairness constraints it is PSPACE-hard in the size of the game structure (Alur, Henzinger, and Kupferman 2002, Theorem 5.5), that is, doubly exponential in the number of variables. In the next section, we restrict our attention to some cases where we can check if the formula is satisfiable more efficiently.

## Linear Execution Strategy

Rather than determining which action has to be applied in each (reachable) state, it might be practical to consider (sequential) plans where the agent applies actions one by one. This leads us to the notion of *linear execution strategies* that are special cases of execution strategies that, roughly speaking, resemble sequential plans (as in classical planning).

Given a sequence of actions that includes $switch_a$, we call the sequence of real actions (all actions except the switch actions) its *non-switch sequence*. Using this notion we can define a linear execution strategy as follows:

**Definition 9.** *Let* $\mathcal{P} = (V, A, E, I, G)$ *be a planning task,* $\mathcal{S} = \langle P, S, F, \pi, d, \delta \rangle$ *be its concurrent game structure, and let* $\epsilon$ *be an execution strategy. If every outcome of* $\epsilon$ *shares the same non-switch sequence of actions* $\theta$*, then* $\epsilon$ *is a* **linear execution strategy***. We call* $\theta$ *the* **action sequence** *of* $\epsilon$.

The following example shows that linear execution strategies are indeed more limited than general execution strategies. Consider a variant of the AUV example, where a ship drops an object onto a random place that the AUV wants to collect. A general execution strategy of the AUV will wait until the ship drops the object and then will act to collect the object. However, there is no linear execution strategy that can solve this problem, as the AUV cannot commit to a specific sequence of actions beforehand. On the other hand, if the location of the object is fixed at the beginning, the AUV can commit to a sequence of actions and hence can use a linear execution strategy to solve the problem.

In order to verify if a linear execution strategy is valid, we have to investigate what nature can do and whether it can apply a sequence of events that might jeopardize the applicability of the next action or the goal rendering the agent's linear execution strategy unsuccessful. Hence we define an *alive* property determining whether a (partial) variable assignment has always a chance to become true despite acts of nature (while considering the fairness assumption).

**Definition 10.** *Let* $\mathcal{P} = (V, A, E, I, G)$ *be a planning task and* $S$ *be the set of states over* $V$*. We define a* **nature game** *of* $\mathcal{P}$ *as a labelled transition system* $\mathcal{N} = (S, T)$*, where* $(s, e, s') \in T$ *iff* $\gamma(s, e) = s'$.

*We say that* $s'$ *(resp. an event* $e \in E$*) is* **reachable** *from* $s$ *(in* $\mathcal{N}$*) iff there exists a path from* $s$ *to* $s'$ *(resp. a state* $s''$ *such that* $s'' \models pre(e)$*) in* $\mathcal{N}$*. Otherwise, we say that* $s'$ *(resp.* $e$*) is* **unreachable** *from* $s$ *(in* $\mathcal{N}$*).*

*Let* $q$ *be a (partial or complete) variable assignment over* $V$*. We say that* $q$ *is* **alive** *with respect to* $s$ *(in* $\mathcal{N}$*), denoted as* $alive(q, s)$*, iff for each state* $s'$ *reachable from* $s$ *there exists a state* $s''$ *with* $s'' \models q$ *that is reachable from* $s'$.

For convenience, we define $\Delta : 2^S \times A \rightarrow 2^S$ as $\Delta(S', a) = \{s \mid s' \in S', \gamma(s, a) = s'\}$ representing regression from $S'$ via $a$, that is, what are the states where applying action $a$ results in some state from $S'$.

**Theorem 1.** *Let* $\mathcal{P} = (V, A, E, I, G)$ *be a planning task, let* $S$ *be the set of states over* $V$ *and* $\mathcal{N}$ *be the nature game of* $\mathcal{P}$*. Finally, let* $\theta = \langle a_1, \ldots, a_n \rangle$ *be an action sequence. Then, we specify sets of states* $S^0, S^1, \ldots, S^n$ *as follows.*
- $S^n = \{s \mid s \in S, s \models ha(a_n), alive(G, s)\}$
- $S^i = \{s \mid s \in S, s \models ha(a_i), \exists s' \in \Delta(S^{i+1}, a_{i+1}) : alive(s', s)\}$ *for all* $1 \leq i < n$
- $S^0 = \{s \mid s \in S, \exists s' \in \Delta(S^1, a_1) : alive(s', s)\}$

*If* $I \in S^0$*, then there exists a valid linear execution strategy* $\epsilon_\theta$ *for* $\mathcal{P}$ *and* $S^0, \ldots S^n \neq \emptyset$.

*Proof.* The intuition behind the proof is to show that we can regressively construct sets of states that ensure the (eventual) applicability of the subsequent non-switch actions or achievability of the goal (after the agent applies all non-switch actions). It can be seen that for a state $s$ and a variable assignment $q$ such that $alive(q, s)$ it holds that $\mathcal{S}_{switch}, s \models \langle\langle agent \rangle\rangle \diamond q$, where $\mathcal{S}_{switch}$ is a variant of concurrent game

structure from Definition 5 that allows only the *switch*$_a$ action for the agent, as nature cannot generate a sequence of events making states entailing $q$ unreachable.

For a state $s_n \in S^n$, we can derive that $\mathcal{S}_{switch}, s_n \models \langle\langle agent \rangle\rangle \diamond G$ and that $s_n$ can be achieved just after $a_n$ is applied by the agent (because of $s_n \models ha(a_n)$). Then, we can observe that if the agent applies $a_n$ in any state from $\Delta(S^n, a_n)$ it eventually achieves the goal $G$. For a state $s_i \in S^i$ ($0 \leq i < n$), we can similarly derive that $\mathcal{S}_{switch}, s_i \models \langle\langle agent \rangle\rangle \diamond (\bigvee_{s' \in \Delta(S^{i+1}, a_{i+1})} s')$ and that, if $i \geq 1$, then $s_i$ can be achieved just after $a_i$ is applied by the agent (because of $s_i \models ha(a_i)$). We can then observe that if the agent applies $a_i$ in any state from $\Delta(S^i, a_i)$, it eventually reaches a state from $\Delta(S^{i+1}, a_{i+1})$ and eventually achieves the goal $G$ (by induction to $S^n$). From this observation, we can immediately see that if $I \in S^0$, then we can define a valid linear execution strategy $\epsilon_\theta$ which follows $\theta$. Also, if $S^i = \emptyset$, we can derive that $S^j = \emptyset$ for every $0 \leq j < i$ and hence if $I \in S^0$, then $S^0, \ldots S^n \neq \emptyset$. $\qquad \square$

The definition of linear execution strategy (Definition 9) is, however, not constructive as it contains an implicit ambiguity for the agent in deciding when to apply the next action. To make the definition constructive we leverage the concept of *wait-for preconditions* that represents a fragment of *social laws* in multi-agent planning (Karpas, Shleyfman, and Tennenholtz 2017). In our case, wait-for preconditions are sets of states that uniquely define when the agent applies its non-switch action and when it switches.

In the AUV example, the AUV might have to wait until a ship passes through the location before entering. This is a stronger condition than the *move* action requires; however, if the AUV enters the location before the ship passes it, the ship might run over the AUV.

**Definition 11.** *Let $\mathcal{P} = (V, A, E, I, G)$ be a planning task and $\mathcal{S} = \langle D, S, F, \pi, d, \delta \rangle$ be its concurrent game structure. Let $\theta_w = \langle (w(a_1), a_1), \ldots, (w(a_n), a_n) \rangle$ be a sequence of actions (from $A$) associated with wait-for preconditions $w(a_i) \subseteq S (1 \leq i \leq n)$ and $\epsilon_{\theta_w}$ be a linear execution strategy. If for every outcome of $\epsilon_{\theta_w}$, it is the case that non-switch actions are applied in states satisfying actions' wait-for preconditions, i.e., $s \in w(a_i)$, while switch actions are applied in states in which the wait-for precondition for the next action is not met, then $\epsilon_{\theta_w}$ is a **linear execution strategy with wait-for preconditions**.*

The following theorem, which follows immediately from Theorem 1 and Definition 11, shows how wait-for preconditions can be refined from $S^i$ states.

**Theorem 2.** *Let $\mathcal{P} = (V, A, E, I, G)$ be a planning task, $\epsilon_\theta$ be a linear execution strategy whose action sequence is $\theta = \langle a_1, \ldots, a_n \rangle$ and sets of states $S^0, S^1, \ldots, S^n$. Then, we can refine $\epsilon_{\theta_w}$, a linear execution strategy with wait-for preconditions, by computing wait-for precondition, for each action $a_i \in \theta$, as $w(a_i) = \Delta(S^i, a_i)$.*

## Verification of Linear Execution Strategies

Theorems 1 and 2 provide a blueprint of how action sequences can be verified as linear execution strategies (with wait-for preconditions). An important step of the verification is to guarantee that the next action (or the goal) will become eventually applicable regardless of how the nature acts (if it follows the fairness assumption). For that reason, we have to identify whether a (partial) variable assignment is "alive" with respect to a state (see Definition 10) can be done by leveraging the notion of *strongly connected component*, which is well known in the graph theory, and the related notion of *condensation* of a graph, where each strongly connected component is "condensed" to a single node. The idea is to compute strongly connected components and "condense" the nature game to get an understanding of its topology. That is important in determining the alive relation for (partial) variable assignments.

**Theorem 3.** *Let $\mathcal{P} = (V, A, E, I, G)$ be a planning task and $\mathcal{N} = (A, T)$ be its nature game. Let $N^1, \ldots, N^k$ be sets of nodes forming strongly connected components of $\mathcal{N}$. For some (partial) variable assignment $q$ and a state $s$, it is the case that $alive(q, s)$ if and only if for $N^i$ such that $s \in N^i$ it is the case that there does not exist a path from $N^i$ to some (condensed) leaf node $N^j$ such that $\nexists s' \in N^j : s' \models q$.*

*Proof.* If $alive(q, s)$, then for every $s''$ reachable from $s$ in $\mathcal{N}$ it holds that $s'$ such that $s' \models q$ is reachable from $s''$ in $\mathcal{N}$ (according to Definition 10). Since the condensation of $\mathcal{N}$ is acyclic there is a path from $N^i$ to at least one (condensed) node (including $N^i$ itself). We can also observe that $s'$ (with $s' \models q$) has to be in a (condensed) leaf node, otherwise, there might exist $s''$ in $\mathcal{N}$ from which $s'$ might not be reachable. If, however, there is a path from $N^i$ ($s \in N^i$) to another (condensed) leaf node $N^j$ such that $\nexists s' \in N^j : s' \models q$, then we can find such $s''$ (being, for example, in $N^j$) from which $s'$ (such that $s' \models q$) is not reachable, and hence $alive(q, s)$ would not hold. $\qquad \square$

**Corollary 1.** *Let $\mathcal{P} = (V, A, E, I, G)$ be a planning task, $\mathcal{N} = (A, T)$ be its nature game, and $N^1, \ldots, N^k$ be sets of nodes forming strongly connected components of $\mathcal{N}$. Let $q$ be a (partial) variable assignment. If $N^i$ is a (condensed) leaf node such that $s \in N^i$ and $s \models q$, then for each $s' \in N^i$ it is the case that $alive(q, s')$. If, on the other hand, there does not exist a (condensed) leaf node $N^i$ such that $s \in N^i$ and $s \models q$, then $alive(q, s')$ does not hold for any state $s'$.*

To simplify reasoning about the alive relations we propose a special case of a *Domain Transition Graph (DTG)* (Jonsson and Bäckström 1998) that considers how the values of variables can be changed in the nature game.

**Definition 12.** *Let $\mathcal{P} = (V, A, E, I, G)$ be a planning task. For each $v \in V$, we define the **Nature Domain Transition Graph (NDTG)** as a directed graph $\mathcal{G}^v = (D(v), T^v)$, where $D(v)$ is a set of nodes and $T^v$ set of edges such that for all $x, y \in D(v)$ with $x \neq y$ and $e \in E$, $(x, y) \in T^v$ iff $eff(e)[v] = y$ and either $pre(e)[v] = x$ or $v \notin vars(pre(e))$. Also, we denote $x \rightarrow_v y$ if there is a path from $x$ to $y$ in $\mathcal{G}^v$, $x \nrightarrow_v y$ if not, and $\downarrow_v x$ if $x$ is a leaf node in $\mathcal{G}^v$.*

## Maintaining the Value of a Variable

Firstly, we consider three situations in which a given value of the variable is maintained (permanently, or eventually).

By a simple analysis of NDTG we can identify that the alive relation holds for values in leaf nodes as those values cannot be modified by events in any reachable state in the nature game. In the AUV example, we can observe that, for example, the position of the AUV is always maintained regardless of event occurrence.

**Lemma 1.** *Let $\mathcal{P} = (V, A, E, I, G)$ be a planning task, $S$ be the set of states over $V$, $v \in V$ be a variable and $\mathcal{G}^v = (D(v), T^v)$ be its NDTG. If for $x \in D(v)$ it holds that $\downarrow_v x$, then for each state $s \in S$ such that $s[v] = x$ it is the case that $alive((v, x), s)$*

In a more general sense, we can also observe that the value for a given variable does not change if none of the events deleting it is reachable from a given state. In the AUV example, the fact determining whether the AUV is operational cannot be deleted if the AUV is not in the ship's corridor, or the ship has already passed by.

**Lemma 2.** *Let $\mathcal{P} = (V, A, E, I, G)$ be a planning task, $S$ be the set of states over $V$, and $v \in V$ be a variable. For $s \in S$ it holds that $alive((v, x), s)$ if $s[v] = x$, and for every $e \in E$ deleting $(v, x)$ it holds that it is unreachable from $s$.*

Another case considers situations in which an event might delete a value of a variable but some other event that can eventually occur can reachieve that value. In a modified AUV example, in which the ship cannot run over the AUV, we can observe that the ship might temporarily block a cell (i.e., it deletes the fact that the cell is free) but will eventually move away and make the cell again free.

**Lemma 3.** *Let $\mathcal{P} = (V, A, E, I, G)$ be a planning task, $S$ be the set of states over $V$ and $v \in V$ be a variable. For $s \in S$ it is the case that $alive((v, x), s)$ if $s[v] = x$ and for every $e \in E$ applicable in $s$ deleting $(v, x)$, there exists an event $e' \in E$ such that $ha(e) \models pre(e')$, $eff(e')[v] = x$ and for every event $e'' \in E$ reachable from $\gamma(s, e)$ and deleting some precondition of $e'$ it holds that either $eff(e'')[v] = x$ or $pre(e'')[v] = x$.*

*Proof.* It can be seen that if any event $e$ changes the value of $v$ from $x$ to some other value, there can eventually occur another event $e'$ that reverts the value of $v$ back to $x$. Also, any event $e''$ that can possibly interfere with the applicability of $e'$ either requires or achieves $(v, x)$. Hence, $alive((v, x), s)$ for each $s$ with $s[v] = x$. $\square$

Note that even though two facts over different variables can be determined as alive according to Lemma 3 their conjunction might not be alive as, for example, an event $e$ deletes $(v, x)$ and achieves $(v', x')$ and another event $e'$ does it the other way round.

## Connecting Different Variable Values

If the value of a given variable has to be changed, we can investigate whether nature can eventually apply events that achieve the required value, and such a value is then maintained. NDTG can be analyzed in the sense of achieving a leaf value of a variable from another value. It can be possible if events on the path from that value to the leaf one

---

**Algorithm 1: Verifying a Linear Execution Strategy**

**Require:** A planning task $\mathcal{P} = (V, A, E, I, G)$, a sequence of actions $\epsilon = \langle a_1, a_2, \ldots, a_n \rangle$
**Ensure:** A linear execution strategy with wait-for preconditions over $\theta = \langle (w(a_1), a_1), \ldots, (w(a_n), a_n) \rangle$ being a solution of $\mathcal{P}$

1: $\theta \leftarrow \langle \rangle$; $s \leftarrow G$
2: **for** $i \leftarrow n, i \geq 0, i - -$ **do**
3:     $s_p = ha(a_i)$ if $i \geq 1$, or $s_p = I$ otherwise
4:     $pre_w(a_i) \leftarrow pre(a_i)$
5:     **while** $\exists v \in (vars(s_p) \cap vars(s)) : s_p[v] \neq s[v]$ **do**
6:         **if** $\exists \langle e_1, \ldots, e_k \rangle$ as in Lemma 4 **then**
7:             **for** $v' \in vars(pre(e_1)) \cap vars(pre_w(a_i))$ : $s_p[v'] = pre_w(a_i)[v'] \leftarrow pre(e_1)[v']$
8:             $s_p \leftarrow \gamma(s_p \langle e_1, \ldots, e_k \rangle)$
9:         **else**
10:             **return** Fail
11:     **for all** $v \in (vars(s_p) \cap vars(s))$ **do**
12:         **if** none of Lemmas 1–3 can be applied **then**
13:             **return** Fail
14:         **else**
15:             $pre_w(a_i)$.add(Cond-Vals$(s_p, a_i, e, v)$)
16:     $s \leftarrow Reg(s, a_i)$; $\theta$.push-back$(pre_w(a_i), a_i))$
17: **return** $\theta$

---

can eventually occur, which is formally stated in the following lemma. In the AUV example, we might observe that the ship will eventually leave the area, i.e., it will be changing its position until it (finally) leaves the area.

**Lemma 4.** *Let $\mathcal{P} = (V, A, E, I, G)$ be a planning task, $S$ be the set of states over $V$ and $v \in V$ be a variable and $\mathcal{G}^v = (D(v), T^v)$ be its NDTG. For $s \in S$ with $s[v] = y$ it is the case that $alive((v, x), s)$ if (i) there exists a path $y = q_0, q_1, \ldots, q_k = x$ in $\mathcal{G}^v$ (i.e., $y \to_v x$), (ii) $\downarrow_v x$ (iii) there exists a sequence of events $\langle e_1, \ldots, e_k \rangle$ such that for every $1 \leq i \leq k$ it holds that $pre(e_i)[v] = q_{i-1}$, $eff(e_i)[v] = q_i$, $\gamma(s, \langle e_1, \ldots, e_{i-1} \rangle) \models pre(e_i)$ and for each $e'_i$ that deletes a fact required by $e_i$ it is the case that $e'_i$ is unreachable or $v \in vars(pre(e'_i))$ and $pre(e'_i)[v] \neq q_{i-1}$.*

*Proof.* The sequence of events $\langle e_1, \ldots, e_k \rangle$ from the assumption can eventually achieve the value $x$ of the variable $v$ from $y$. On top of that, the fact $(v, x)$ cannot be deleted because of (ii). In particular, it is assured that each event (from that sequence) can be eventually applied because of $\gamma(s, \langle e_1, \ldots, e_{i-1} \rangle) \models pre(e_i)$ and the fact that any event possibly invalidating the precondition of any of the events in the sequence is either unreachable or requires a different value of $v$ than that being currently set. $\square$

## The Method

"To verify" sequences of actions are generated by off-the-shelf classical planners such that a planning task $\mathcal{P} = (V, A, E, I, G)$ is converted into a classical planning task $\mathcal{P}_c = (V, A \cup E, I, G)$, solved, and from the solution of $\mathcal{P}_c$, denoted as $\epsilon_c$, which contain both agent's actions and nature's events, we take out events, i.e., $\epsilon = \epsilon_c \setminus E$.

Such an action sequence $\epsilon$ has to be verified whether it can yield a valid linear execution strategy for $\mathcal{P}$. The verification of $\epsilon$ can be done regressively step by step as indicated in Theorem 1 and, consequently, wait-for preconditions can be computed for the actions from $\epsilon$ as indicated in Theorem 2. Note that we abuse the notation by considering wait-for preconditions ($pre_w(a)$) as partial states (or partial variable assignments) meaning that $pre_w(a)$ represents all states in which $pre_w(a)$ holds. To determine the regression step (over partial) states we define the $\text{Reg}(s, a)$ function that is calculated according to (Pommerening and Helmert 2015), i.e., $\text{Reg}(s, a)$ is defined only on variables from $(vars(s) \setminus vars(eff(a))) \cup vars(pre_w(a)))$ such that $\text{Reg}(s,a)[v] = pre_w(a)[v]$ if $v \in vars(pre_w(a))$, or $\text{Reg}(s,a)[v] = s[v]$ otherwise. Even though Theorem 3 gives a blueprint on how the alive relation, which is a necessary element of the verification process, can be computed, it requires enumerating (almost) the whole state space. Hence, we leverage Lemmas 1 to 4 to determine some alive relations in polynomial time (if sequences of events satisfying Lemma 4 are generated greedily). Although such a simplification compromises the completeness of the verification approach, it allows us to leverage classical planners without a large overhead to generate linear execution strategies in a subclass of scenarios.

The verification algorithm is summarised in Algorithm 1. We start in the partial state containing only the goal facts and iteratively regress through the plan to the initial state. In an intermediate step, we look for whether we can, for all relevant variables, claim the alive relation from a partial state ($s_p$), determined by either $ha(a_i)$ or the initial state (after we processed all actions of the sequence), to the current partial state ($s$). At first, we process variables whose values differ in $s$ and $s_p$ by leveraging Lemma 4 (Line 6). If we can find a sequence of events satisfying the lemma, we may need to update $pre_w(a_i)$ by considering extra preconditions needed to ensure applicability of the event sequence (because of the condition (iii) of Lemma 4, we need to consider only the precondition of the first event, i.e., $pre(e_1)$), and we also update $s_p$ reflecting that the sequence of events has been applied (Line 7). If in any case, we fail to apply Lemma 4, then we conclude that the verification has failed. Then, we process variables whose values are the same in $s$ and $s_p$. We leverage Lemmas 1–3 (note that Lemma 3 can be applied at most once in the $i$-th step) and if none of them can be applied, we conclude that the verification has failed. If we use Lemma 2, then the "Cond-vals" function works as follows (for other lemmas it returns an empty variable assignment). For each not unreachable event $e$ deleting the respective value of $v$ we try to invalidate its precondition by looking for another variable that is not (yet) considered in $s_p$ (and neither in $pre_w(a_i)$) such that we can find a leaf node in the variable's NDTG having a different value than the value of the variable in $pre(e)$. If we find such a variable and its value, we add it into $pre_w(a_i)$ (Line 15). Note that the unreachability checks that are part of some of the lemmas are done on the abstraction level, i.e., by checking for the non-existence of paths in NDTGs of respective variables.

| Domain | Type | 1 | 2 | 3 | 4 | 5 |
|--------|------|------|-------|------|------|------|
| AUV | VLES | 0.08 | 0.09 | 0.10 | 0.11 | 0.13 |
| AUV | FOND-1 | 9.56 | 39.15 | - | - | - |
| HR | VLES | 0.06 | 0.07 | 0.07 | 0.09 | 0.12 |
| HR | FOND-1 | 15.48 | - | - | - | - |

Table 1: Runtime results (in s) on the AUV and HR domains.

## Experimental Evaluation

Our experiments aim to demonstrate the potential of our method for Verifying Linear Execution Strategies (VLES) in terms of scalability despite possibly large non-deterministic branching caused by actions of nature. To give a perspective, we compared our VLES method with a method based on FOND planning that considers that nature can apply at most one event in its turn (FOND-1) (Chrpa, Pilát, and Gemrot 2019), which is an easier problem to solve (as we consider infinite sequences of events nature can apply in its turn).

For the comparison, we use the AUV domain introduced by Chrpa, Gemrot, and Pilát (2020), where an AUV (controlled by the agent) has to collect resources in a grid environment in which there are ships (controlled by nature) passing through in their designed corridors (columns of the grid). If a ship enters the cell with the AUV, then the AUV is destroyed. In our case, we consider that each ship can pass through the area only once. We designed 5 problems ranging from 4x4 to 8x8 grid size, 4 to 8 resources, and 1 to 5 ships. We have also designed a HomeRobot (HR) domain that involves a robot (controlled by the agent) that needs to make up rooms. Rooms are connected by a corridor, so to move between rooms one has to enter the corridor first. There are also humans (controlled by nature) that can move between rooms as well. The corridor, however, has limited space and at most one entity can be there at the same time. We designed 5 problems ranging from 4 to 12 rooms, and 2 to 6 humans.

For generating plans as an input to VLES (i.e., solving classical planning problems considering both actions and events and then removing events from the plans), we used LAMA (Richter and Westphal 2010) and for solving FOND-1 problems we used PRP (Muise, McIlraith, and Beck 2012). The time limit for each problem was 900 seconds and the memory limit was 4GB. The experiments were run on AMD Ryzen 5 5500u 2.1GHz, 16GB RAM, Ubuntu 22.04.[1]

Table 1 shows the runtime comparison of VLES and FOND-1 approaches (note that both planning and verification runtimes are included in the VLES case). The results show that VLES scales reasonably well despite the increase in the number of events nature can apply. Note that FOND-1 results are shown to demonstrate how detrimental impact on performance the non-deterministic branching can have (FOND-1 considers a milder assumption than planning against nature does) rather than to make a direct comparison against VLES, which is incomplete in general. Nevertheless, the results of VLES indicate that focusing on linear execution strategies in planning against nature has good potential despite the incompleteness of such an approach.

---

[1]Our source code and benchmarks will be provided in CRC.

## Related Work

The concept of exogenous events in planning (Dean and Wellman 1990) was used in systems such as Circa (Musliner, Durfee, and Shin 1993). These systems usually have to reason with a whole (or almost whole) state space. Markov Decision Process (MDP)-based approaches can be leveraged to tackle events (Mausam and Kolobov 2012) and aim to generate a policy with the most promising action in each state. Monte-Carlo Tree Search (MCTS) approaches provide similar benefits; however, the success rate tends to drop for problems with dead-ends (Patra et al. 2021).

Alternatively, classical planning techniques can be used to generate plans without explicitly considering events and if an event occurs during the plan execution and changes the state of the environment to an unknown one, then a new plan is generated (Komenda, Novák, and Pechoucek 2014). The success of FF-replan (Yoon, Fern, and Givan 2007) in the International Planning Competition 2006 (it was an unofficial winner of the probabilistic track) indicates that planning against nature tasks can be addressed by interleaving (classical) planning, plan execution, and re-planning if the agent is in an unexpected state or cannot execute the following action. However, in domains with dead-ends such an approach might not be effective (and might even be dangerous).

To address the issue of encountering dead-ends while planning and acting in the environment with exogenous events using classical planning Chrpa, Gemrot, and Pilát (2020) adapted the notion of *safe states* (Cserna et al. 2018), where a state is safe if no sequence of events can transform it to a dead-end state. If such an event sequence exists, the state is unsafe. The main idea is to generate *robust plans* connecting one safe state to another. Robust plans are guaranteed to always succeed despite event occurrence. The main drawback of the technique is that it tries to find robust plans between safe states online which might not always be possible. If there is no way of transiting an unsafe area via a robust plan, the agent gets stuck forever (albeit in a safe state). A subsequent work of Chrpa, Pilát, and Med (2021) presents a technique for generating *eventually applicable plans* that, in our terminology, refer to linear execution policies for problems with events. That technique determines "cyclic phenomena" that are formed by reversible events and also identifies potentially irreversible events that might lead to dead-ends. These irreversible events cannot occur during plan execution which is guaranteed, informally speaking, by ensuring that such events cannot become applicable during plan execution (the agent either does not enable them or disables them before they have a chance to occur). It should be noted that works of Chrpa, Gemrot, and Pilát (2020) and Chrpa, Pilát, and Med (2021) consider a different execution model in which an action of the agent is interleaved by a set of independent events (might be empty). Our work, on the other hand, considers valid sequences of events between actions of the agent. Hence concepts from Chrpa et al.'s works cannot be directly applied in our model.

### Planning Against Nature and FOND Planning

Fully Observable Non-deterministic (FOND) Planning, in a nutshell, concerns tasks in which the environment is fully observable while actions have several different outcomes and if one such action is applied a random outcome occurs (Cimatti et al. 2003). The task is to find a strong plan that, in the context of our terminology, represents a valid execution strategy that is a solution of a FOND planning task. For instance, the well-known PRP planner (Muise, McIlraith, and Beck 2012) looks for strong plans by leveraging classical planning techniques and handling non-determinism by attempting to "close" states from which there does not yet exist a plan. FOND planning is known to be EXPTIME-complete (Littman, Goldsmith, and Mundhenk 1998).

Although FOND planning and planning against nature share some aspects such as non-determinism and full environment observability, there is a fundamental difference in how non-determinism occurs. In FOND planning, non-determinism is triggered only by (non-deterministic) actions of the agent while in planning against nature non-determinism is triggered by events that nature can apply. That means, that the number of non-deterministic alternatives occurring after the agent applies an action $a$ in a state $s$ is the number of reachable states from $\gamma(s, a)$ in the corresponding nature game. In the AUV example, ships can move freely regardless of the movement of the AUV, and after the AUV moves, each of the ships can then move to any reachable position or stay, so the number of non-deterministic alternatives corresponds to the number of combinations of reachable positions of the ships. Hence, in each "turn" the number of outcomes of nature might be exponential with respect to the size of the representation of the planning task (against nature). We conjecture that planning against nature is computationally harder than FOND planning and at most double exponential (as ATL model checking).

## Conclusion

In this paper, we have formalized the problem of *planning against nature* as a concurrent game structure and how to tackle it by using ATL model checking (Alur, Henzinger, and Kupferman 2002). We then focused on linear execution strategies resembling sequential plans and have shown that if actions in a linear execution strategy are enriched with wait-for preconditions, the strategy then uniquely specifies when the agent has to apply a given action and when it has to wait. We have shown under which circumstances a linear execution strategy is valid (i.e., guarantees eventual goal achievement) and how wait-for preconditions can be extracted. We have then proposed a method for verifying sequential plans that also computes wait-for preconditions. Although the method is incomplete and works on a subclass of problems, we have experimentally shown that focusing on linear execution strategies in planning against nature has the potential to alleviate the high computational demand.

In the future, we plan to investigate how we can effectively generalize the abstraction approach (e.g. by leveraging ideas such as Merge and Shrink (Sievers and Helmert 2021)) and how to extract more complex social laws that would help with generating action sequences forming the basis of linear execution strategies. We also plan to combine linear execution strategy generation and verification into a generate-and-test loop that would cover a larger class of problems.

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
