# OpenReview forum: "On Verifying Linear Execution Strategies in Planning Against Nature"
_icaps-conference.org/ICAPS/2024/Conference — ICAPS 2024_

### Official Review · Reviewer_3ei7 · 2024-01-14

**Significance And Importance:** 2
**Soundness:** 2
**Novelty:** 3
**Clarity:** 3
**Overall Evaluation:** 1
**Confidence:** 4

**Weaknesses:**

0: Minor weaknesses requiring some work to be addressed for the paper to be accepted.

**Contributions Of The Paper:**

Using Alternating-time Temporal Logic, the authors formalise a game between a planning agent and 'nature'; the latter traditionally called the 'environment' in the ATL literature. Nature begins with control, and the two agents take it in turns to act -- i.e., actions do not execute concurrently and are not durative. It is supposed that nature acts fairly, in a formal sense according to constraints (Definition 4), and it seems the planning agent gets arbitrarily many turns at acting, explicitly scheduling an action to pass control to nature. The details regarding how the system evolves after the goal is achieved (for the authors algorithm, after finite agent actions) do not seem important, and we sensibly have an infinite temporal logic semantics to accommodate nature.

The authors suppose that solving the general game against nature is too computationally hard, and focus their attention on games which might have a solution that corresponds to a "linear execution strategy". Here, a candidate solution corresponds to a sequence of planning agent actions, which might be validated (Algorithm 1) and annotated with "wait for" conditions that describe when the planning agent's actions are interleaved/scheduled wrt nature's actions. Algorithm 1 is sound, but incomplete. The overall method developed proceeds first by solving a classical planning problem, where a planner chooses both the planning agent and nature's actions, and then uses the verification algorithm (Algorithm 1) to either derive a solution strategy from that (e.g., by adding wait-for, etc.), or fail to do so.

Experimentation is on ten instances from 2 domains, one an AUV problem from the literature, and the other a household cleaning robot domain that I understand is of the authors' own design. A comparison is made with a FOND planner, which unsurprisingly does not scale to medium sized problems due to branching factors. Given the compared systems solve different problems---e.g., the authors' system solves their own conformant(sort of, but with arbitrarily long wait-fors)-like sequential planning problem---it is not clear what the purpose or value of the experimentation is. Especially given the conditions required for the wait-for algorithm to return a strategy successfully (such as, here nature doesn't repeat itself), the comparison with conformant planning seems relevant.

Given the focus and contributions here are mostly about linear execution strategies, I am surprised that the related work section and experimentation does not consider the conformant planning problem, or algorithms for solving that. The gap between the setting here and FOND seems, to me, comparable to the gap between this setting and conformant planning. Thus, experimentation of the type exhibited here and related work should consider this as an equal point of comparison to FOND.

The ATL formalism admits multiple agents, beyond 2, and action concurrency. These are interesting aspects to investigate in future work, which are not discussed in the future work section. I see this as an interesting strength of the front matter, although this strength seems to be lost once we get to algorithms and conditions for those to be successful.

**Ethical Considerations:**

(1) Not Applicable: The paper does not have any ethical considerations to address

**Nomination For Best Paper:**

No

**Questions For Authors:**

I have no questions at this stage.

**Reproducibility:**

4: Authors promise to release code and domains (whichever apply).

**Strengths Of The Paper:**

The algorithm and solution concept is novel, interesting, and consistent with the claims. Novelty of the problem setting in the context of a new algorithm are the main strengths. Those contributions are substantial and detailed, hence my positive evaluation.

Experimentation indicates the authors have made an interesting first step in addressing their conception of planning against nature. I do not believe the authors made any serious mistakes. The paper is understandable. Given some of the authors data/instances, I expect the experimental results along the lines presented could be reproduced. I did not find any catastrophic errors in the lemmas or theorems which would invalidate the core contribution. More precision and detail regarding fairness would be helpful.

**Weaknesses Of The Paper:**

- The paper would be stronger if there was a real-world problem described motivating: (i) the algorithm, (ii)  this multi-agent formalism that supports concurrency and scenarios beyond 2 agents, or (iii) a range of ATL formulae that might characterise different compelling solution concepts. The authors invoke the ATL literature, but here the contributions (Theorems, Lemmas, and Algorithm) are not obviously benefiting from that.

 - The limited experimentation is a significant weakness.

 - Paper would be stronger with the relationship with conformant planning identified. If applicable comparisons with SOTA conformant planners on benchmarks should be provided, or a deep discussion regarding expected performance should be provided.

 - Definition 4 the constraint given, I believe called a _strong_ fairness constraint in the literature, could be explained better and examples given. The specific fairness constraints that preside in the authors' setting could be given explicitly. It is not obvious, for example, how this definition can be used to ensure that, given the opportunity according to a 'concurrent game structure', nature exhibits all possibilities in an infinite trace.

 - \lambda is overloaded in Definition 4, because after Line 185 it is defined as a sequence of states.

 - Proposition 1 does not say much if nature never gives The (e.g., planning) Agent a turn. Doesn't this break the 'iff' of the proposition.

 - There are multiple distinct logics here, and only one entailment notation. That is not ideal.

 - Def 12, concept of a 'leaf' node of a _directed_ graph is oftentimes defined, however 'sink' node is a more intuitive term.

 - Alg 1, Line 11, typesetting of 'var s'. Line 16, parenthesis are wrong.

 - Definition 1, use 'the set', rather than 'a set of propositions', as the definition I believe intends this to capture _all_ the propositions that are true. Also, this definition allows for partial observability, however that is not a feature of the setting/domains under consideration. Is observability somehow important to your underlying project?

 - The language between multi-valued state variables, the term 'delete' used wrt deleting an assignment to a multi-value variable, and the relation between state variables and logic could be setup better. This would be a very challenging paper to read in the case of a reader who does not already have familiarity with a range of planning and logic concepts. It could be made more accessible to the full ICAPS audience and range of expertise.

---

> ### Author Rebuttal · Authors · 2024-01-25
>
> Thank you for your review. We will take all of the comments into account when preparing the final version.
>
> Indeed, a linear execution strategy is similar to a conformant plan, although they are used in different settings. However, our use of waitfors produces something that is slightly more expressive, as a waitfor condition can be interpreted as a policy with a loop that waits until the waitfor condition is satisfied. We will clarify this point in the final version.

---

### Official Review · Reviewer_JM1x · 2024-01-22

**Significance And Importance:** 2
**Soundness:** 2
**Novelty:** 3
**Clarity:** 3
**Confidence:** 3

**Weaknesses:**

0: Minor weaknesses requiring some work to be addressed for the paper to be accepted.

**Contributions Of The Paper:**

The paper investigates linear execution strategies in planning against nature, where nature controls some of the actions and a linear execution strategy is a strategy that is independent of nature's actions and hence corresponds to a classical plan, augmented with wait-for conditions, which allow the agent to wait for a certain state to occur. Based on a game-theoritcal characterization in the logic ATL, sufficient conditions for the existence of valid linear strategies are characterized and the construction of such strategies including the wait-for conditions is described. Additionally, the paper presents several incomplete but efficient methods for determining such execution strategies and evaluates those methods in two scenarios, demonstrating the general feasibility.

**Ethical Considerations:**

(1) Not Applicable: The paper does not have any ethical considerations to address

**Nomination For Best Paper:**

No

**Overall Evaluation:**

-1: (weak reject)

**Questions For Authors:**

1. Have you evaluated additional domains, possibly where none of the conditions were satisfied?

**Reproducibility:**

4: Authors promise to release code and domains (whichever apply).

**Strengths Of The Paper:**

The paper provides a theoretical characterization and several arguably more practical conditions for valid linear execution strategies, both being non-trivial contributions. The theoretical results seem to be correct and the paper is generally well-written.

**Weaknesses Of The Paper:**

The sufficient conditions for the existence of valid strategies (i.e., Lemma 1-4) are somewhat simple and it is unclear how generally applicable they are, i.e., whether they are applicable in only very simple domains. The experimental evaluation is limited and the two evaluation domains are similar. Therefore, the evaluation also does provide much insight into the general applicability of the approach. The comparison to FOND is somewhat unfair, as the FOND approach is complete (although restricted to single nature events).

---

> ### Author Rebuttal · Authors · 2024-01-25
>
> Thank you for your review. We will take all of the comments into account when preparing the final version, but we address only the main points here.
>
> As Reviewer 3ei7 notes, linear execution policies are similar to conformant plans (see our response to Reviewer 3ei7 for more details on the comparison). Thus, similarly to the way conformant plans can be used inside a contingent planner, in future work, we intend to use our linear plans inside a planner which can solve a richer set of problems. Thus, we believe our contribution is an important building block for further research into planning against nature.
>
> Note that FOND-1 approach is not complete with regards to planning against nature (as FOND-1 only admits at most one event between agent's actions instead of sequences of events).
>
> Q1: We have experimented with problems in which the verification failed, for instance, if the ship in the AUV domain could move back and forth in a grid column making a permanent threat to the AUV that had to cross the column. In such a case, the AUV cannot safely cross the column and hence any plan crossing that column cannot be a linear execution strategy. Runtime-wise, since our verification method runs in polynomial time, the results are similar to the successful cases. We can include the "negative" results in the final version.

---

### Official Review · Reviewer_xW9a · 2024-01-22

**Significance And Importance:** 2
**Soundness:** 3
**Novelty:** 3
**Clarity:** 3
**Overall Evaluation:** 2
**Confidence:** 4

**Weaknesses:**

1: Minor weaknesses that are easily fixable.

**Contributions Of The Paper:**

The paper formalizes how a linear plan can be transformed into an execution strategy that will be valid in nature using "wait-for-preconditions". In nature, this is especially relevant where there are nondeterministic events impacting the outcomes of the actions taken by the actor. The paper models the actor in nature as a concurrent (asynchronous) game using alternating time temporal logic.

The authors define the properties of the plans to be executed in nature and prove these properties under specific sets of assumptions.

Finally, in a small set experimental setup, the authors show that their verification strategy works in two toy domains.

**Ethical Considerations:**

(5) Excellent: The paper comprehensively addresses all of the applicable ethical considerations

**Nomination For Best Paper:**

No

**Questions For Authors:**

(1) What impact will a multi-agent scenario have on the generation and verification of the linear execution strategies?

(2) Does Table 1 only measure the runtime of successful cases, or are failure cases included in it?

______________________________________________
POST REBUTTAL
______________________________________________
Thank you for providing your response. I request the authors to include the requested information in the paper.

After going over the other reviews, I agree with the point that if the authors are comparing with FOND, then a comparison with conformant planning should be done, both experimentally and in discussion of the related work. If it is not feasible to do experimentally, a discussion should be included in the related work.

**Reproducibility:**

4: Authors promise to release code and domains (whichever apply).

**Strengths Of The Paper:**

The paper addresses a very relevant problem in the integration of acting and planning, and provides a way to transform classical planning techniques into linear execution strategies.

The concepts introduced, for example, wait-for-conditions, valid, alive, etc are intuitive and motivated well. Most of the main definitions have an introductory paragraph explaining the idea which is a critical requirement for theoretical papers.

**Weaknesses Of The Paper:**

The lemmas 1-4 are hard to follow. Although this is a theoretically rigorous paper, I don't think that a series of theorems and lemmas is the right way to present the material. To a reader, a more easier way to grasp the material and the concepts quickly would be via a running example which remains consistent throughout the paper. I would suggest using the AUV domain as the main running example. The authors indeed do that in some of the definitions but not all. Some of the theorems and lemmas may be moved to an appendix to make space for more examples. Another suggestion I have is to add an unified figure with the AUV domain illustrating the main definitions.

The list of contributions may be converted into an itemized list to improve readability.

The assumptions made throughout the paper are hard to find.

The experimental evaluation is a bit weak. One of the domains, HR does not have any dead ends.

Minor issues:
1) Algorithm 1: left paran missing in the ensure condition
2) Table 1: Missing column labels for HR domain

---

> ### Author Rebuttal · Authors · 2024-01-25
>
> Thank you for your review. We will take all of the comments into account when preparing the final version.
>
> Q1: If we assume centralized planning for multiple agents producing a sequence of actions (involving actions of all agents), then our approach can be straightforwardly used. For multiple nature agents, our approach can also be straightforwardly used as their events still need to form valid sequences. A decentralized multiagent approach would have to be augmented with a coordination mechanism such as "social laws".
>
> Q2: Table 1 contains successful runs. We have experimented with problems in which the verification fails (more details are provided in the answer to Reviewer JM1x), where the runtimes are similar to those for successful runs (as our verification method runs in polynomial time). We can include the "negative" results in the final version.

---

### Meta-Review · Area_Chair_MrLF · 2024-02-05

**Recommendation:** Accept (Poster)
**Confidence:** 3

**Metareview:**

The reviewers agree that the paper has merits wrt novelty and motivation, that the reported ideas are interesting, and that the line of research is worth pursuing. Yet, the work exhibits some weaknesses, some of which can be reasonably addressed in the camera ready (if the paper is accepted) and others that appear harder to solve, considering also the lack of a further review round. In particular, the reviewers raised concerns about the experimental evaluation, which appears somewhat limited, as: 1. carried out on similar domains and with positive instances only; 2. considering only a comparison with FOND, without discussing the link with conformant planning. In addition, 3. issues were raised about the theoretical results (Lemmas 1-4), which should be explained better, in particular as regards their general applicability.
As to 1., the authors have provided a satisfactory response in their rebuttal, providing details about experiments carried out on negative instances, that can be included in the final version.  As to 3., the raised concerns seem addressable. 2., instead, requires a substantial effort.

Overall, the submitted material appears publishable, assuming the authors fully address 1. and 3., and address 2. by including at least a substantial discussion that positions the work clearly wrt conformant planning.

**Ethical Considerations:**

(1) Not Applicable: The paper does not have any ethical considerations to address